# Real versus Sham Manual Therapy in Addition to Therapeutic Exercise in the Treatment of Non-Specific Shoulder Pain: A Randomized Controlled Trial

**DOI:** 10.3390/jcm11154395

**Published:** 2022-07-28

**Authors:** Fermin Naranjo-Cinto, Adriana-Imelda Cerón-Cordero, Claudia Figueroa-Padilla, Dulce Galindo-Paz, Samuel Fernández-Carnero, Tomás Gallego-Izquierdo, Susana Nuñez-Nagy, Daniel Pecos-Martín

**Affiliations:** 1Universidad de Alcalá, Facultad de Enfermería y Fisioterapia, Departamento de Fisioterapia, Grupo de Investigación en Fisioterapia y Dolor, 28801 Alcalá de Henares, Spain; ft.ferminnaranjo@hotmail.com (F.N.-C.); tomas.gallego@uah.es (T.G.-I.); susana.nunez@uah.es (S.N.-N.); daniel.pecos@uah.es (D.P.-M.); 2Benemérita Universidad Autónoma de Puebla, Facultad de Medicina Licenciatura en Fisioterapia, Puebla 72000, Mexico; adrianaceron170@gmail.com (A.-I.C.-C.); ftclaudiafigueroa@gmail.com (C.F.-P.); dulc.galindo@gmail.com (D.G.-P.)

**Keywords:** shoulder, rotator cuff, musculoskeletal manipulations, exercise

## Abstract

The aim of this study was to evaluate if manual therapy added to a therapeutic exercise program produced greater improvements than a sham manual therapy added to same exercise program in patients with non-specific shoulder pain. This was an evaluator-blinded randomized controlled trial. Forty-five subjects were randomly allocated into one of three groups: manual therapy (glenohumeral mobilization technique and rib-cage technique); thoracic sham manual therapy (glenohumeral mobilization technique and rib-cage sham technique); or sham manual therapy (sham glenohumeral mobilization technique and rib-cage sham technique). All groups also received a therapeutic exercise program. Pain intensity, disability and pain-free active shoulder range of motion were measured post treatment and at 4-week and 12-week follow-ups. Mixed-model analyses of variance and post hoc pairwise comparisons with Bonferroni corrections were constructed for the analysis of the outcome measures. All groups reported improved pain intensity, disability and pain-free active shoulder range of motion. However, there were no between-group differences in these outcome measures. The addition of the manual therapy techniques applied in the present study to a therapeutic exercise protocol did not seem to add benefits to the management of subjects with non-specific shoulder pain.

## 1. Introduction

Non-specific shoulder pain (NSSP) is the most common disorder of the shoulder [1]. The first choice of treatment for NSSP is physiotherapy [2]. The best physiotherapy treatment choice for NSSP is therapeutic exercise. Other treatments such as taping [3], electrotherapy [4] and manual therapy [5] may be proposed as coadjutants. 

Manual therapy has been shown to produce benefits at short-term follow-ups in various musculoskeletal disorders [6]. Manual therapy is thought to produce a neurophysiological response that activates the descending inhibitory pathways, leading to a reduction in the symptomatology of patients [7,8]. Previous studies have evaluated the effects of manual therapy in NSSP; these were mostly thoracic spine manipulation, glenohumeral and cervical spine mobilizations and soft tissue techniques [9]. However, most of them only evaluated the effects at the post-treatment follow-up [10,11,12], thus hindering knowledge of the possible benefits of these therapies over a longer time [9]. In the last decades, several authors have designed sham manual therapy procedures to compare with real manual therapy techniques with the aim of controlling the confounding factors; this has improved the quality of the manual therapy research field [13].

The hypothesis of the present study was that the addition of manual therapy to a therapeutic exercise program would produce better benefits in comparison with the same exercise program with sham manual therapy procedures in the management of patients with NSSP.

## 2. Materials and Methods

### 2.1. Study Design

This randomized controlled trial was conducted according to the recommendations of the Consolidated Standards of Reporting Trials (CONSORT) statement [14].

The study protocol was registered in the database of the System for the Registration of Research Protocols of Health Services from the Estado de Puebla (registration number 590). Ethical approval was obtained from the Ethical Committee of Benemérita Universidad Autónoma de Puebla, Mexico (2/111/729) and a clinicatrials.gov registration was obtained (number NCT04440046). The study was conducted according to the Declaration of Helsinki.

### 2.2. Subjects

A convenience sample of subjects with NSSP was recruited through announcements at the Benemérita Universidad Autónoma de Puebla, Mexico. All subjects signed a consent form before participating in the study.

The inclusion criteria were: age between 18 and 60 years; unilateral shoulder pain of a non-traumatic origin lasting more than 3 months; pain evoked with active shoulder movements; pain and weakness evoked with manual muscle strength tests for abduction and/or external rotation; and minimal or no rest pain (less than 3 points on a numeric pain rating scale).

The exclusion criteria were: other shoulder pathologies other than NSSP; systemic diseases; neural symptoms; neck pain; radiculopathy; or treated for shoulder pain in the previous three months.

Demographic data about age, sex, height, weight, body mass index, dominant side and painful side were also collected. 

### 2.3. Sample Size

The sample size calculation was based on the time-by-group interaction of a three-by-four mixed-model analysis of variance (ANOVA). The effect size was estimated to be 0.25 with a correlation between the repeated measures of 0.50, a sphericity correction of 0.75, 80% power and an α value of 0.05. According to the sample size calculation, 45 subjects (15 per group) had to be recruited.

### 2.4. Measurements

All measurements were carried out at Benemérita Universidad Autónoma de Puebla, Mexico. The subjects were measured at the baseline and post-treatment follow-up as well as at 4-week and 12-week follow-ups after treatment. All measurements were obtained by a physiotherapist with more than five years of experience who was unaware of the treatment group allocation.

#### 2.4.1. Pain Intensity

The pain intensity during the previous week (primary outcome) was measured with a visual analog scale (VAS), where 0 was no pain and 10 was the worst pain imaginable. The VAS has shown a good reliability with an intraclass correlation coefficient (ICC) value ranging from 0.71 to 0.99 [15,16].

#### 2.4.2. Shoulder Pain and Disability Index

The degree of shoulder disability was measured with a Shoulder Pain and Disability Index (SPADI) questionnaire that was transculturally adapted from English into Spanish in 2015 [17]. The SPADI scores range from 0 (no disability) to 130 (maximum degree of disability). The final score of the questionnaire was reported as a percentage from the maximum possible score. 

#### 2.4.3. Range of Movement

Shoulder range of movement was measured for flexion, extension, abduction and internal and external rotations. It was measured with a two-arm goniometer (Baseline^®^, USA). The maximum pain-free active range of motion was recorded for each movement. Three measurements were taken for each movement and the mean was used for the statistical analysis. The goniometer has shown a good reliability for the measurement of shoulder range of movement in previous studies (ICC 0.91 to 0.99) [18].

Shoulder flexion and abduction were measured with the subject in a standing position. Shoulder extension was measured with the subject in a prone position with the elbow flexed to 90 degrees outside the stretcher. Shoulder internal rotation was measured with the subject in a prone position with the shoulder at 90 degrees of abduction and the elbow flexed to 90 degrees. Finally, shoulder external rotation was measured with the subject in a supine position with the shoulder at 90 degrees of abduction and the elbow flexed to 90 degrees [18].

### 2.5. Interventions

Epidat 4.2 software (version 4.2, julio 2016. Consellería de Sanidade, Xunta de Galicia, España) was used to randomly allocate the subjects into treatment groups. The allocation was concealed in sealed envelopes that were sequentially numbered. All groups received two treatment sessions per week for five weeks. Both the researcher and the participants were blinded to the allocation. 

All groups received a therapeutic exercise program. One group was treated with a sham manual therapy directed to the shoulder and thoracic spine. Another group was treated with a real manual therapy directed to the shoulder and a sham manual therapy directed to the thoracic spine. The final group was treated with real manual therapy directed to the shoulder and thoracic spine.

The treatments were performed by three physiotherapists with more than five years of experience in manual therapy.

#### 2.5.1. Therapeutic Exercise

The therapeutic exercise program consisted of isometric exercises with a progressive load [19]. The isometric exercises were self-resisted by the patient in a sitting position and the exercises were performed for shoulder flexion, abduction, internal rotation and external rotation. A total of 3 repetitions per exercise were performed, with 20 s of contraction in each repetition and a 10 s rest period in between them. The patient was instructed to perform the exercises every day and to increase the load based on their tolerance to the exercises.

#### 2.5.2. Real Manual Therapy

For the glenohumeral mobilization technique, the patient laid in a supine position with their arm alongside their body with 30 degrees of shoulder abduction and 90 degrees of elbow flexion. The physiotherapist placed one hand at the distal third of the arm and the other one at the proximal third of the forearm. The therapist induced rhythmic tractions to the glenohumeral head with their arm/hand whilst inducing a flexion–extension movement to the elbow with their forearm/hand at the same time. When traction was applied, the elbow was moved toward flexion and when traction was released, the elbow was moved toward extension. This technique is a type of neural sliding mobilization technique. A total of 3 series of 15 repetitions at a frequency of 2 Hz were performed.

For the rib-cage technique, the patient laid in a prone position with both arms relaxed alongside their body. The physiotherapist placed both thumbs above the ipsilateral second rib near the costotransverse joint. The therapist performed a posterior to anterior rhythmic mobilization over 3 min at a frequency of 2 Hz.

#### 2.5.3. Sham Manual Therapy

The sham manual therapy was conducted in the same position as the real manual therapy. The physiotherapist placed their hands in the same way as with the real manual therapy without inducing any type of movement, but maintained the contact with their hands on the skin of the patient for 3 min.

The treatment within the intervention groups was the same; however, the pain of the participants was taken into consideration, carrying out the treatment without the presence of pain.

No adverse effects were recorded in any of the treatment groups.

### 2.6. Statistical Analysis

The data normality was evaluated with a Shapiro–Wilk test. For the descriptive analysis of the quantitative variables, the mean and standard deviation (SD) were reported. For the categorical variables, the absolute frequencies and percentages were calculated. The homogeneity of the groups was evaluated with a one-way ANOVA for the quantitative variables and a chi-squared test for the categorical variables [20].

An intention-to-treat analysis was used for assessing the differences in the outcome measures. Three-by-four mixed-model ANOVAs were constructed with the time (baseline, post treatment, 4 weeks and 12 weeks) as the within-subjects factor and the group (sham, sham thoracic and real manual therapy) as the between-subjects factor. Post hoc pairwise comparisons were conducted with a Student’s *t*-test with a Bonferroni correction. The effect size of the main effects and the interaction of the ANOVA were estimated with a partial eta squared (η_p_^2^) [20].

All the analyses were conducted with SPSS V.22 software (SPSS Inc., Chicago, IL, USA). All the analyses were performed assuming an α level of 0.05 with 95% confidence intervals (CIs).

## 3. Results

The final sample was composed of 45 subjects (with no dropouts at the follow-up), with 15 subjects per group. The sham group had a mean age of 30.93 years (SD: 10.87), the thoracic sham group had a mean age of 36 years (SD: 15.70) and the real manual therapy group had a mean age of 35.73 years (SD: 13.66). The demographic characteristics of the subjects are presented in Table 1.

### 3.1. Pain and Disability

The three-by-four mixed ANOVA revealed a significant main effect of time for pain (F = 24.33; *p* < 0.01; η_p_^2^ = 0.65) and disability (F = 33.82; *p* < 0.01; η_p_^2^ = 0.72). There was a non-significant main effect for the group and a non-significant time-by-group interaction for pain and disability. The post hoc comparisons are presented in Table 2.

### 3.2. Range of Movement

The three-by-four mixed ANOVA revealed a significant main effect of time for range of movement in flexion (F = 25.82; *p* < 0.01; η_p_^2^ = 0.66), extension (F = 7.25; *p* < 0.01; η_p_^2^ = 0.15), abduction (F = 14.67; *p* < 0.01; η_p_^2^ = 0.26), internal rotation (F = 5.12; *p* < 0.01; η_p_^2^ = 0.12) and external rotation (F = 15.43; *p* < 0.01; η_p_^2^ = 0.27). There was a non-significant main effect of the group and a non-significant time-by-group interaction for all ranges of movement as seen in Table 3.

## 4. Discussion

There was a statistically significant decrease in pain intensity at the 4-week and 12-week follow-ups, but not at the post-treatment follow-up. Disability measured with the SPADI showed a statistically significant decrease at all follow-ups. There were no between-group differences.

The absence of a decrease in pain intensity at the post-treatment follow-up disagrees with the results of previous studies [10]. However, most of these previous published studies evaluated pain intensity at rest or during movement and/or orthopedic tests whereas in the present study, pain intensity was measured as the mean intensity during the previous week, which could explain these discrepancies. Furthermore, the differences found by previous studies were near to the minimum detectable change (MDC) of 1.1 points [21]. Previous studies have focused mostly on thoracic spine manipulation techniques [9,10] whereas the techniques of the present study have not previously been used in randomized controlled trials, which could explain the observed discrepancies. The pain improvements at the 4-week and 12-week follow-ups were above the MDC of 1.1 points [21] and the decrease at 12 weeks was also above the MDC of 2 points, as reported by other studies [15,22].

Although there was a statistically significant improvement in disability measured with the SPADI at the post-treatment follow-up, this improvement did not exceed the threshold of the MDC of 12.2 [17]. The decrease in the SPADI at the 4-week follow-up was near the MDC (mean difference: −11.60) and the improvement at the 12-week follow-up exceeded the MDC (mean difference: −18.73). These results agreed with previous studies that found that improvements in shoulder function attained with exercise were greater at the 12-week follow-up [23].

The absence of between-group differences in the present study agreed with the results of other previous studies [5,10], which suggests that the improvements observed were due to therapeutic exercise. However, the other previous investigations that have used different types of manual therapy techniques other than articular manipulations or mobilizations found an improvement associated with these techniques [9], thus hindering any conclusions about the benefits of the addition of manual therapy to a therapeutic exercise program for pain intensity and disability.

There was a statistically significant improvement in all movements over a longer time without between-group differences. Only extension movement showed a statistically significant improvement at the post-treatment follow-up; however, this increase did not exceed the lower limit of the MDC of 6–11° reported for shoulder goniometry measurements [18]. At the 4-week follow-up, the improvements observed were statistically significant for all movements, except flexion. However, these improvements were near to the lower limit of the MDC of 6°, with only external rotation exceeding it (mean difference: 7.53°). At the 12-week follow-up, all movements showed a significant increase; the most important ones were observed for flexion (mean difference: 10.47°) and abduction (mean difference: 11.64°), which exceeded the upper limit of the MDC of 11° [18]. There are only a few published studies that have evaluated pain-free active range of movement as an outcome and most of them only measured it post treatment or at short-term follow-ups [18,24,25]. The results of the present study suggest that therapeutic exercise could increase pain-free active range of movement of the shoulder in all directions, especially flexion and abduction, at the 12-week follow-up, but manual therapy does not seem to add any benefit to this improvement in subjects with NSSP.

## 5. Limitations

Due to the absence of a control group without a treatment and the fact that the efficacy of blinding was not assessed, we could not ensure that the observed improvements were not due to the natural history of NSSP or a placebo effect.

The use of two techniques of manual therapy not evaluated in any previous randomized controlled trial made it difficult to provide a comparison of the present research with previous published studies. Although no significant between-group differences in the outcome measures were found, other techniques of manual therapy could add benefits to a program of therapeutic exercise in patients with NSSP.

Due to the inherent characteristics of the participants, the results obtained here should be evaluated with caution and not be transferred to other population groups.

The variability of the pathologies associated with NSSP was not taken into account when the selecting criteria were applied.

## 6. Conclusions

The addition of the two manual therapy techniques used in the present study to a therapeutic exercise program did not seem to add benefits to the management of subjects with NSSP. Further research is needed to evaluate other types of techniques of manual therapy at mid-term and long-term follow-ups to clarify the role of manual therapy in the management of NSSP.

## Figures and Tables

**Table 1 jcm-11-04395-t001:** Baseline characteristics of the subjects (*n* = 45).

Characteristic, Mean (SD)	Sham (*n* = 15)	Thoracic Sham (*n* = 15)	Real MT (*n* = 15)	*p*-Value
Age, years	30.93 (10.87)	36 (15.70)	35.73 (13.66)	0.52
Height, cm	169.07 (10.29)	165.33 (9.82)	165.86 (11.06)	0.54
Weight, kg	75 (15.81)	74.84 (16.06)	70.25 (13.11)	0.55
BMI, kg/m^2^	26.12 (4.43)	27.57 (6.76)	25.41 (3.01)	0.46
Sex, *n* (%)				0.08
Male	11 (26.7)	7 (46.7)	5 (33.3)	
Female	4 (73.3)	8 (53.3)	10 (66.7)	
Dominant side, *n* (%)				1.00
Right	14 (93.3)	14 (93.3)	15 (100)	
Left	1 (6.7)	1 (6.7)	0 (0)	
Painful side, *n* (%)				0.74
Right	10 (66.7)	11 (73.3)	9 (60)	
Left	5 (33.3)	4 (26.7)	6 (40)	

SD: standard deviation; MT: manual therapy; BMI: body mass index.

**Table 2 jcm-11-04395-t002:** Pain and disability differences from the baseline.

Variable	Baseline			Post Treatment			4 Weeks			12 Weeks		
	Sham	Thoracic Sham	Real MT	Sham	Thoracic Sham	Real MT	Sham	Thoracic Sham	Real MT	Sham	Thoracic Sham	Real MT
Outcomes, mean (SD)												
VAS, cm	3.48 (1.95)	3.71 (1.82)	3.44 (1.33)	2.68 (1.57)	2.82 (1.95)	3.26 (1.86)	1.69 (1.46)	1.67 (1.47)	1.46 (1.27)	1.23 (1.33)	1.05 (1.35)	0.69 (1.28)
SPADI	25.73 (17.49)	28.00 (13.42)	31.13 (19.44)	20.73 (18.84)	21.33 (14.76)	24.93 (20.46)	20.47 (18.97)	13.33 (15.32)	16.27 (19.02)	13.13 (15.07)	5.93 (7.99)	9.60 (15.36)
Overall group differences from baseline, mean (95% CI)												
VAS, cm				−0.62 (−1.26, 0.01)			−1.94 ^‡^ (−2.75, −1.13)			−2.56 ^‡^ (−3.38, −1.73)		
SPADI				−5.96 ^‡^ (−10.24, −1.68)			−11.60 ^‡^ (−19.30, −3.90)			−18.73 ^‡^ (−25.01, −12.45)		

Statistically significant (*p* < 0.05); ^‡^ statistically significant (*p* < 0.01). MT: manual therapy; SD: standard deviation; VAS: visual analog scale; SPADI: shoulder pain and disability index; CI: confidence interval.

**Table 3 jcm-11-04395-t003:** Range of movement differences from the baseline.

Variable	Baseline			Post Treatment			4 Weeks			12 Weeks		
	Sham	Thoracic Sham	Real MT	Sham	Thoracic Sham	Real MT	Sham	Thoracic Sham	Real MT	Sham	Thoracic Sham	Real MT
Outcomes, mean (SD)												
Pain-free range of movement, degrees												
Flexion	156.87 (5.41)	153.73 (11.03)	149.80 (18.63)	144.73 (18.08)	144.26 (27.69)	149.34 (24.70)	160.93 (10.27)	152.73 (21.76)	158.53 (18.33)	163.73 (6.39)	165.93 (9.92)	162.13 (17.17)
Extension	30.87 (6.45)	28.77 (10.33)	31.47 (8.37)	32.73 (7.99)	35.05 (9.90)	34.43 (10.59)	35.80 (7.18)	35.87 (7.95)	34.40 (8.24)	36.67 (8.34)	36.93 (7.38)	34.53 (8.10)
Abduction	145.27 (22.87)	152.73 (20.31)	149.20 (22.67)	141.15 (26.48)	153.89 (20.31)	149.35 (23.35)	149.33 (25.02)	159.40 (18.77)	156.33 (22.70)	154.93 (22.32)	169.00 (12.94)	158.20 (23.12)
I. Rotation	41.21 (13.42)	52.07 (18.24)	48.60 (10.56)	46.50 (15.00)	54.13 (12.71)	51.06 (13.94)	50.13 (11.21)	53.13 (14.53)	56.47 (15.00)	49.33 (14.34)	58.73 (14.81)	57.40 (14.65)
E. Rotation	64.47 (18.81)	61.87 (17.38)	61.67 (20.64)	61.81 (17.67)	62.13 (14.74)	66.67 (17.82)	70.60 (20.93)	61.33 (16.39)	68.67 (17.23)	73.93 (17.75)	75.57 (15.26)	67.27 (17.24)
Overall group differences from baseline, mean (95% CI)				Post Treatment			4 Weeks			12 Weeks		
Pain-free range of movement, degrees												
Flexion				−7.36 (−16.56, 1.85)			3.93 (−2.91, 10.78)			10.47 ^‡^ (6.32, 14.62)		
Extension				3.70 ^†^ (0.27, 7.13)			4.99 ^‡^ (1.37, 8.61)			5.68 ^‡^ (1.68, 9.67)		
Abduction				−0.94 (−5.90, 4.03)			5.96 ^†^ (0.73, 11.18)			11.64 ^‡^ (4.85, 18.44)		
I. Rotation				3.27 (−3.05, 9.59)			5.95 ^†^ (0.43, 11.47)			7.86 ^‡^ (1.60, 14.13)		
E. Rotation				0.87 (−3.55, 5.29)			7.53 ^‡^ (2.23, 12.84)			9.59 ^‡^ (4.38, 14.80)		

^†^ Statistically significant (*p* < 0.05); ^‡^ Statistically significant (*p* < 0.01). MT: manual therapy; SD: standard deviation; I: internal; E: external; CI: confidence interval.

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
