# Peer review of "Real versus Sham Manual Therapy in Addition to Therapeutic Exercise in the Treatment of Non-Specific Shoulder Pain: A Randomized Controlled Trial"

_jcm, 2022, doi:10.3390/jcm11154395_

Round 1

Reviewer 1 Report

Overall

Thank you for the opportunity to review this paper. The authors performed a RCT in patients with rotator cuff related shoulder pain. The topic seems to be relevant for the audience of J of Clinical Medicine and the authors have spent a lot of time in this project. However, this study contains some flaws. One of my problems is the content of the manual therapy treatment. Manual therapy consisted of genohumeral joint mobilizations and second rib mobilizations. However, the rationale and evidence for using these techniques is unclear. Moreover, manual therapy is much more than these two techniques and also consists of providing information/ advice, exercise therapy etc etc, not just two isolated techniques. Treatment is not tailored to the participants’ needs and findings from physical examination. Information about blinding of participants or success of blinding of the assessors is missing.

The authors state that only post-treatment effects are evaluated, and that RCTs with longer follow-up periods are needed. Unfortunately, this study did also not measure the long-terms effects, but used a follow-up period of maximum 12 weeks which could not be considered as long- or intermediate term follow-up. Moreover, the selection criteria are not clearly reported (e.g., which tests are used and what is known about the validity for RC related shoulder pain implying the question whether patients with rotator cuff related disorders are included?), and study population is relatively young. This hinders the generalizability.

The English level and style should be improved throughout the manuscript.

 Abstract:

Line 11: Delete ‘real’ manual therapy

Line 13: Please change exercise ‘protocol’ into exercise ‘program’

Line 14: Rephrase sentence and please do not start with a number. The groups are unclear to me. What is sham or real manual therapy? 

What is known

Line 29: This implies that this study is not needed as this also only assesses short-term results (4 and 12 weeks). Twelve weeks is still considered as short-term follow-up.

Introduction

Line 40: The reported mechanisms of action of manual therapy are only able to explain differences in pain, not for other outcomes.

Methods

Line 58: The trial is registered in a local registration system. This makes it hard to check whether the trial was conducted as originally planned.

Line 109 : Are any adverse events recorded?

Line 210: The interventions are briefly presented. I would recommend to use the TIDier checklist to report the interventions

Line 66: Why were participants above the age of 60 excluded? This decreases the generalizability of the results.

Line 80: What is the primary outcome measure of this study?

Line 102: Reliability values from previous studies are used. But how reliable could this physiotherapist measure ROM? These measures are operator-dependent.

Line 111-113: Please, create a separate paragraph about randomization and blinding

Line 158: Intention-to-treat analyses not mentioned.

Line 122: What is known about the effectiveness of this exercise protocol used in this patient population?

Line 128: Manual therapy is much broader than glenohumeral joint mobilizations, and AP pressure on the second rib. Was there a need for rib mobilizations (e.g., did the participants have painful or restricted second ribs or glenohumeral joints)? I lack the rationale for the techniques used. Treatment should be tailored to the findings of the clinical examination and needs of the patients.

Line 148: In studies with low numbers of participants (i.e. n=15), the Shapiro-Wilk will provide not-significant results, (incorrectly) assuming normally distributed data. Visual inspection will give a better insight into the distribution of the data. I am not convinced that alle variables are normally distributed in the manuscript.

Results:

Table 1: This table does not give a good impression of the participant characteristics. Information about clinical characteristics such as pain intensity, comorbidity, psychosocial factors, medication use and the most important prognostic factors are missing.

Discussion

Line 188: Main findings should be the between group differences, not the within group differences. There were no between group differences (main effects for group)

Line 196: For relevant within group differences, we have to compare the results with the minimal clinical important change (MCID) not with the MDT (measurement error)

Line 209-210: This is incorrect. The changes can also be related to the natural course or placebo-effects. The authors needed to include a no treatment group to draw these conclusions.

Line 233: The patient population is relatively young, so the results cannot be extrapolated to other age groups. This should also be reported as a limitation.

Moreover, we lack information about blinding of the participants or assessor. Was the success of blinding measured?

Author Response

Comments and Suggestions for Authors

Overall

Thank you for the opportunity to review this paper. The authors performed a RCT in patients with rotator cuff related shoulder pain. The topic seems to be relevant for the audience of J of Clinical Medicine and the authors have spent a lot of time in this project. However, this study contains some flaws. One of my problems is the content of the manual therapy treatment. Manual therapy consisted of genohumeral joint mobilizations and second rib mobilizations. However, the rationale and evidence for using these techniques is unclear.

Moreover, manual therapy is much more than these two techniques and also consists of providing information/ advice, exercise therapy etc etc, not just two isolated techniques. Treatment is not tailored to the participants’ needs and findings from physical examination. Information about blinding of participants or success of blinding of the assessors is missing.

The authors state that only post-treatment effects are evaluated, and that RCTs with longer follow-up periods are needed. Unfortunately, this study did also not measure the long-terms effects, but used a follow-up period of maximum 12 weeks which could not be considered as long- or intermediate term follow-up. Moreover, the selection criteria are not clearly reported (e.g., which tests are used and what is known about the validity for RC related shoulder pain implying the question whether patients with rotator cuff related disorders are included?), and study population is relatively young. This hinders the generalizability.

The English level and style should be improved throughout the manuscript.

Thanks for the comments, the reply to the previous paragraphs is described separately in each of the points where they are referred to.

 Abstract:

Line 11: Delete ‘real’ manual therapy

            Thanks for the comment, the suggested corrections have been carried out. Line 11.

Line 13: Please change exercise ‘protocol’ into exercise ‘program’

            Thanks for the comment. Suggested change was made. Line 13.

Line 14: Rephrase sentence and please do not start with a number. The groups are unclear to me. What is sham or real manual therapy? 

            Thanks for the comment. Suggested change was made. A clearer description of the study groups was carried out. Lines 14-18.

What is known

Line 29: This implies that this study is not needed as this also only assesses short-term results (4 and 12 weeks). Twelve weeks is still considered as short-term follow-up.

We appreciate the comment. There is evidence to suggest that in the case of RCRSP a follow-up between 12-26 weeks is considered a mid-term follow-up[1,2].

Introduction

Line 40: The reported mechanisms of action of manual therapy are only able to explain differences in pain, not for other outcomes.

Thanks for the comment. There is evidence to suggest that Manual therapy is thought to produce a neurophysiological response that activates descending inhibitory pathways leading to a reduction in pain[1,3], furthermore improved range of motion, strength and function in patients whit shoulder pain.

Methods

Line 58: The trial is registered in a local registration system. This makes it hard to check whether the trial was conducted as originally planned.

Thanks for the comment. The local ethical committee registration was made and the clinical trial was registered on the clinicatrials.gov page too. Line 63.

Line 109 : Are any adverse events recorded?

Line 210: The interventions are briefly presented. I would recommend to use the TIDier checklist to report the interventions

We appreciate the recommendation, we have carried out the TIDieR checklist, confirming that the description of the treatment has all the requested points.

Line 66: Why were participants above the age of 60 excluded? This decreases the generalizability of the results.

Line 80: What is the primary outcome measure of this study?

The main outcome is the pain intensity. Line 90

Line 102: Reliability values from previous studies are used. But how reliable could this physiotherapist measure ROM? These measures are operator-dependent.

Thanks for the comment. The goniometer has shown good reliability for the measurement of shoulder range of movement in previous studies (ICC, 0.91 to 0.99) and standard error measurement 2°-3° in range of motion[4]. Line 106.

Line 111-113: Please, create a separate paragraph about randomization and blinding

Thanks for the comment. Suggested change was made. Line 115.

Line 158: Intention-to-treat analyses not mentioned.

Thanks for the comment, the suggested corrections have been carried out. Line 164.

Line 122: What is known about the effectiveness of this exercise protocol used in this patient population?

            The effectivnees of this kind of protocol has been demostrated in a systematic review and meta-analysis of randomized controlled trials, however, there is no evidence where the same protocol has been used[5].

Line 128: Manual therapy is much broader than glenohumeral joint mobilizations, and AP pressure on the second rib. Was there a need for rib mobilizations (e.g., did the participants have painful or restricted second ribs or glenohumeral joints)? I lack the rationale for the techniques used. Treatment should be tailored to the findings of the clinical examination and needs of the patients.

Based on the existing scientific evidence[6,7], the research team assumes that the benefit obtained from manual therapy is related to systemic neurophysiological responses that lead to pain inhibition, which is considered the main variable to treat in patients with shoulder pain related to the rotator cuff. This is the reason why we assume that regardless of the characteristics of each participant, the manual therapy protocol carried out could improve the symptoms of each patient.

Line 148: In studies with low numbers of participants (i.e. n=15), the Shapiro-Wilk will provide not-significant results, (incorrectly) assuming normally distributed data. Visual inspection will give a better insight into the distribution of the data. I am not convinced that alle variables are normally distributed in the manuscript.

Thank for the comment. Must be taken into account that sample size is described in lines 80-83. Since the full sample size was achieved, the result presented are in line with the methodology designed.

Results:

Table 1: This table does not give a good impression of the participant characteristics. Information about clinical characteristics such as pain intensity, comorbidity, psychosocial factors, medication use and the most important prognostic factors are missing.

Thanks for the comment. Table two shows the evolution in the intensity of pain at the different measurement times, nevertheless information according to comorbility and psychosocial factors was no considered.

Discussion

Line 188: Main findings should be the between group differences, not the within group differences. There were no between group differences (main effects for group).

Thanks for the comment. The results were compared between the groups and no statistically significant differences were found. Lines 199-200.

Line 196: For relevant within group differences, we have to compare the results with the minimal clinical important change (MCID) not with the MDT (measurement error).

Thanks for the comment. The hypothesis described in lines 53-55 attempts to measure the sum of treatments thus the outcome results of these consists of measuring the minimal detectable changes, this is the reason why we decided to this this kind of statistical analyses.

Line 209-210: This is incorrect. The changes can also be related to the natural course or placebo-effects. The authors needed to include a no treatment group to draw these conclusions.

Thanks for the comment. The possibility that the changes are due to the natural course or placebo-effects is considered and described as a limitation in line 244-246

Line 233: The patient population is relatively young, so the results cannot be extrapolated to other age groups. This should also be reported as a limitation.

Thanks for the comment. This has been described as a limitation of the study. Lines 252-253

Moreover, we lack information about blinding of the participants or assessor. Was the success of blinding measured?

Thanks for the comment. We have added information about the blinding. Line 118.The effectiveness of blinding of participants was not assessed, we will describe as a limitation of the study. Line 243-244. 

REFERENCES

  1. Cook T, Lowe CM, Maybury M, Lewis JS. Are corticosteroid injections more beneficial than anaesthetic injections alone in the management of rotator cuff-related shoulder pain? A systematic review. Br J Sports Med [Internet]. 2018 Apr 1 [cited 2022 Jun 24];52(8):497–504. doi:10.1136/BJSPORTS-2016-097444
  2. Cook T, Lewis J. Rotator Cuff-Related Shoulder Pain: To Inject or Not to Inject? J Orthop Sports Phys Ther [Internet]. 2019 May 1 [cited 2022 Jun 24];49(5):289–93. doi:10.2519/JOSPT.2019.0607
  3. Bang MD, Deyle GD. Comparison of supervised exercise with and without manual physical therapy for patients with shoulder impingement syndrome. J Orthop Sports Phys Ther [Internet]. 2000 [cited 2022 Jun 24];30(3):126–37. doi:10.2519/JOSPT.2000.30.3.126
  4. Kolber MJ, Fuller C, Marshall J, Wright A, Hanney WJ. The reliability and concurrent validity of scapular plane shoulder elevation measurements using a digital inclinometer and goniometer. Physiother Theory Pract. 2012 Feb;28(2):161–8. doi:10.3109/09593985.2011.574203
  5. Naunton J, Street G, Littlewood C, Haines T, Malliaras P. Effectiveness of progressive and resisted and non-progressive or non-resisted exercise in rotator cuff related shoulder pain: a systematic review and meta-analysis of randomized controlled trials. Clin Rehabil [Internet]. 2020 Sep 1 [cited 2022 Jun 24];34(9):1198–216. doi:10.1177/0269215520934147
  6. Bialosky JE, Beneciuk JM, Bishop MD, Coronado RA, Penza CW, Simon CB, et al. Unraveling the mechanisms of manual therapy: Modeling an approach. Vol. 48, Journal of Orthopaedic and Sports Physical Therapy. Movement Science Media; 2018. p. 8–18. doi:10.2519/jospt.2018.7476
  7. Desjardins-Charbonneau A, Roy JS, Dionne CE, Frémont P, Macdermid JC, Desmeules F. The efficacy of manual therapy for rotator cuff tendinopathy: a systematic review and meta-analysis. J Orthop Sports Phys Ther [Internet]. 2015 May 1 [cited 2022 Jun 24];45(5):330–50. doi:10.2519/JOSPT.2015.5455

Reviewer 2 Report

The present study could not clarify the positive effect of manual therapy on the management of RCRSP. However, the reviewer believes that negative results also contribute to the accumulation of scientific knowledge.  However, the reviewer will suggest some minor corrections.

The biggest problem in this paper is that the definition of RCRSP is ambiguous. The RCRSP includes subacromial impingement syndrome, rotator cuff tendinopathy, and symptomatic partial and full-thickness rotator cuff tears. Thus, the authors should clarify the distribution of shoulder pathology. If the authors cannot clarify shoulder pathology because of the lack of adequate image examination, including plain roentgenogram or magnetic resonance imaging, the authors describe it in the limitation.

Abstract

The abstract was well written

Introduction

This section was appropriately written to clarify problems regarding the management of RCRSP, and the amount of the sentence was appropriate.

Materials and Methods

Line 66-70: The authors should clarify the distribution of shoulder pathology, which causes RCRSP.

Results

Results were presented with appropriate tables and were easy to understand.

Discussion

The discussion was well documented and clearly relevant to the conclusions.

Limitations

If the authors could not clarify the distribution of shoulder pathology, please add it as a limitation.

 Conclusion

This section was appropriately written.

Author Response

Comments and Suggestions for Authors

The present study could not clarify the positive effect of manual therapy on the management of RCRSP. However, the reviewer believes that negative results also contribute to the accumulation of scientific knowledge.  However, the reviewer will suggest some minor corrections.

The biggest problem in this paper is that the definition of RCRSP is ambiguous. The RCRSP includes subacromial impingement syndrome, rotator cuff tendinopathy, and symptomatic partial and full-thickness rotator cuff tears. Thus, the authors should clarify the distribution of shoulder pathology. If the authors cannot clarify shoulder pathology because of the lack of adequate image examination, including plain roentgenogram or magnetic resonance imaging, the authors describe it in the limitation.

Thanks for the comment. It has been included in limitations. Lines 253-254.

Abstract

The abstract was well written

Introduction

This section was appropriately written to clarify problems regarding the management of RCRSP, and the amount of the sentence was appropriate.

Materials and Methods

Line 66-70: The authors should clarify the distribution of shoulder pathology, which causes RCRSP.

 Thanks for the comment. Cause It has not been clarified, this has been described in limitations. Lines 254-255.

Results

Results were presented with appropriate tables and were easy to understand.

Discussion

The discussion was well documented and clearly relevant to the conclusions.

Limitations

If the authors could not clarify the distribution of shoulder pathology, please add it as a limitation.

Thanks for the comment. It has been included in limitations. Lines 254-255.

 Conclusion

This section was appropriately written.

Round 2

Reviewer 1 Report

Unfortunately, the authors did not respond to my main concern of this study. The rationale and evidence for using the manual therapy techniques are still unclear and not reported. It is insufficient to refer generic neurophysiological responses. Glenohumeral joint mobilizations and second rib mobilizations cannot be considered as ‘real manual therapy’. The therapy is also not adapted to the findings from the patient interview and physical examination. Important biopsychosocial factors were not collected. The clinical relevance and novelty of this study is therefore lacking. Also the sample size is small.

 The authors did also not answer all the specific comments, and only minor changes  to the manuscript are made. 

Author Response

Dear reviewer

Attached you can find the main document with all the corrections made and track-changes active. This points were sent previously but it looks something went wrong.

Also detailed information about the changes done, can be read here with the references supported at the end:

("See responses highlighted in yellow")

Overall

Thank you for the opportunity to review this paper. The authors performed a RCT in patients with rotator cuff related shoulder pain. The topic seems to be relevant for the audience of J of Clinical Medicine and the authors have spent a lot of time in this project. However, this study contains some flaws. One of my problems is the content of the manual therapy treatment. Manual therapy consisted of genohumeral joint mobilizations and second rib mobilizations. However, the rationale and evidence for using these techniques is unclear.

Moreover, manual therapy is much more than these two techniques and also consists of providing information/ advice, exercise therapy etc etc, not just two isolated techniques. Treatment is not tailored to the participants’ needs and findings from physical examination. Information about blinding of participants or success of blinding of the assessors is missing.

The authors state that only post-treatment effects are evaluated, and that RCTs with longer follow-up periods are needed. Unfortunately, this study did also not measure the long-terms effects, but used a follow-up period of maximum 12 weeks which could not be considered as long- or intermediate term follow-up. Moreover, the selection criteria are not clearly reported (e.g., which tests are used and what is known about the validity for RC related shoulder pain implying the question whether patients with rotator cuff related disorders are included?), and study population is relatively young. This hinders the generalizability.

The English level and style should be improved throughout the manuscript.

Thanks for the comments, the reply to the previous paragraphs is described separately in each of the points where they are referred to.

 Abstract:

Line 11: Delete ‘real’ manual therapy

            Thanks for the comment, the suggested corrections have been carried out. Line 11.

Line 13: Please change exercise ‘protocol’ into exercise ‘program’

            Thanks for the comment. Suggested change was made. Line 13.

Line 14: Rephrase sentence and please do not start with a number. The groups are unclear to me. What is sham or real manual therapy? 

            Thanks for the comment. Suggested change was made. A clearer description of the study groups was carried out. Lines 14-18.

What is known

Line 29: This implies that this study is not needed as this also only assesses short-term results (4 and 12 weeks). Twelve weeks is still considered as short-term follow-up.

We appreciate the comment. There is evidence to suggest that in the case of RCRSP a follow-up between 12-26 weeks is considered a mid-term follow-up[1,2].

Introduction

Line 40: The reported mechanisms of action of manual therapy are only able to explain differences in pain, not for other outcomes.

Thanks for the comment. There is evidence to suggest that Manual therapy is thought to produce a neurophysiological response that activates descending inhibitory pathways leading to a reduction in pain[1,3], furthermore improved range of motion, strength and function in patients whit shoulder pain.

Methods

Line 58: The trial is registered in a local registration system. This makes it hard to check whether the trial was conducted as originally planned.

Thanks for the comment. The local ethical committee registration was made and the clinical trial was registered on the clinicatrials.gov page too. Line 63.

Line 109 : Are any adverse events recorded?

Line 210: The interventions are briefly presented. I would recommend to use the TIDier checklist to report the interventions

We appreciate the recommendation, we have carried out the TIDieR checklist, confirming that the description of the treatment has all the requested points.

Line 66: Why were participants above the age of 60 excluded? This decreases the generalizability of the results.

Line 80: What is the primary outcome measure of this study?

Line 102: Reliability values from previous studies are used. But how reliable could this physiotherapist measure ROM? These measures are operator-dependent.

Thanks for the comment. The goniometer has shown good reliability for the measurement of shoulder range of movement in previous studies (ICC, 0.91 to 0.99) and standard error measurement 2°-3° in range of motion[4]. Line 106.

Line 111-113: Please, create a separate paragraph about randomization and blinding

Thanks for the comment. Suggested change was made. Line 115.

Line 158: Intention-to-treat analyses not mentioned.

Thanks for the comment, the suggested corrections have been carried out. Line 164.

Line 122: What is known about the effectiveness of this exercise protocol used in this patient population?

            The effectivnees of this kind of protocol has been demostrated in a systematic review and meta-analysis of randomized controlled trials, however, there is no evidence where the same protocol has been used[5].

Line 128: Manual therapy is much broader than glenohumeral joint mobilizations, and AP pressure on the second rib. Was there a need for rib mobilizations (e.g., did the participants have painful or restricted second ribs or glenohumeral joints)? I lack the rationale for the techniques used. Treatment should be tailored to the findings of the clinical examination and needs of the patients.

Based on the existing scientific evidence[6,7], the research team assumes that the benefit obtained from manual therapy is related to systemic neurophysiological responses that lead to pain inhibition, which is considered the main variable to treat in patients with shoulder pain related to the rotator cuff. This is the reason why we assume that regardless of the characteristics of each participant, the manual therapy protocol carried out could improve the symptoms of each patient.

Line 148: In studies with low numbers of participants (i.e. n=15), the Shapiro-Wilk will provide not-significant results, (incorrectly) assuming normally distributed data. Visual inspection will give a better insight into the distribution of the data. I am not convinced that alle variables are normally distributed in the manuscript.

Thank for the comment. Must be taken into account that sample size is described in lines 80-83. Since the full sample size was achieved, the result presented are in line with the methodology designed.

Results:

Table 1: This table does not give a good impression of the participant characteristics. Information about clinical characteristics such as pain intensity, comorbidity, psychosocial factors, medication use and the most important prognostic factors are missing.

Thanks for the comment. Table two shows the evolution in the intensity of pain at the different measurement times, nevertheless information according to comorbility and psychosocial factors was no considered.

Discussion

Line 188: Main findings should be the between group differences, not the within group differences. There were no between group differences (main effects for group).

Thanks for the comment. The results were compared between the groups and no statistically significant differences were found. Lines 199-200.

Line 196: For relevant within group differences, we have to compare the results with the minimal clinical important change (MCID) not with the MDT (measurement error).

Thanks for the comment. The hypothesis described in lines 53-55 attempts to measure the sum of treatments thus the outcome results of these consists of measuring the minimal detectable changes, this is the reason why we decided to this this kind of statistical analyses.

Line 209-210: This is incorrect. The changes can also be related to the natural course or placebo-effects. The authors needed to include a no treatment group to draw these conclusions.

Thanks for the comment. The possibility that the changes are due to the natural course or placebo-effects is considered and described as a limitation in line 244-246

Line 233: The patient population is relatively young, so the results cannot be extrapolated to other age groups. This should also be reported as a limitation.

Thanks for the comment. This has been described as a limitation of the study. Lines 252-253

Moreover, we lack information about blinding of the participants or assessor. Was the success of blinding measured?

Thanks for the comment. We have added information about the blinding. Line 118.The effectiveness of blinding of participants was not assessed, we will describe as a limitation of the study. Line 243-244.

REFERENCES

  1. Cook T, Lowe CM, Maybury M, Lewis JS. Are corticosteroid injections more beneficial than anaesthetic injections alone in the management of rotator cuff-related shoulder pain? A systematic review. Br J Sports Med [Internet]. 2018 Apr 1 [cited 2022 Jun 24];52(8):497–504. doi:10.1136/BJSPORTS-2016-097444
  2. Cook T, Lewis J. Rotator Cuff-Related Shoulder Pain: To Inject or Not to Inject? J Orthop Sports Phys Ther [Internet]. 2019 May 1 [cited 2022 Jun 24];49(5):289–93. doi:10.2519/JOSPT.2019.0607
  3. Bang MD, Deyle GD. Comparison of supervised exercise with and without manual physical therapy for patients with shoulder impingement syndrome. J Orthop Sports Phys Ther [Internet]. 2000 [cited 2022 Jun 24];30(3):126–37. doi:10.2519/JOSPT.2000.30.3.126
  4. Kolber MJ, Fuller C, Marshall J, Wright A, Hanney WJ. The reliability and concurrent validity of scapular plane shoulder elevation measurements using a digital inclinometer and goniometer. Physiother Theory Pract. 2012 Feb;28(2):161–8. doi:10.3109/09593985.2011.574203
  5. Naunton J, Street G, Littlewood C, Haines T, Malliaras P. Effectiveness of progressive and resisted and non-progressive or non-resisted exercise in rotator cuff related shoulder pain: a systematic review and meta-analysis of randomized controlled trials. Clin Rehabil [Internet]. 2020 Sep 1 [cited 2022 Jun 24];34(9):1198–216. doi:10.1177/0269215520934147
  6. Bialosky JE, Beneciuk JM, Bishop MD, Coronado RA, Penza CW, Simon CB, et al. Unraveling the mechanisms of manual therapy: Modeling an approach. Vol. 48, Journal of Orthopaedic and Sports Physical Therapy. Movement Science Media; 2018. p. 8–18. doi:10.2519/jospt.2018.7476
  7. Desjardins-Charbonneau A, Roy JS, Dionne CE, Frémont P, Macdermid JC, Desmeules F. The efficacy of manual therapy for rotator cuff tendinopathy: a systematic review and meta-analysis. J Orthop Sports Phys Ther [Internet]. 2015 May 1 [cited 2022 Jun 24];45(5):330–50. doi:10.2519/JOSPT.2015.5455
